

# Effects of *in-situ* stroboscopic training on visual, visuomotor and reactive agility in youth volleyball players

Michał Zwierko[1], Wojciech Jedziniak[2], Marek Popowczak[1] and Andrzej Rokita[1]

[1] Department of Team Sports Games, Wroclaw University of Health and Sport Sciences, Wroclaw, Poland
[2] Institute of Physical Culture Sciences, University of Szczecin, Szczecin, Poland

## ABSTRACT

**Background:** Stroboscopic training is based on an exercise with intermittent visual stimuli that force a greater demand on the visuomotor processing for improving performance under normal vision. While the stroboscopic effect is used as an effective tool to improve information processing in general perceptual-cognitive tasks, there is still a lack of research focused on identifying training protocols for sport-specific settings. Therefore, we aimed at assessing the effects of *in-situ* stroboscopic training on visual, visuomotor and reactive agility in young volleyball players.

**Methods:** Fifty young volleyball athletes (26 males and 24 females; mean age, 16.5 ± 0.6 years) participated in this study and were each divided randomly into an experimental group and a control group, who then both performed identical volleyball-specific tasks, with the experimental group under stroboscopic influence. The participants were evaluated three times using laboratory based tests for simple and complex reaction speed, sensory sensitivity and saccade dynamics; before the after the 6-week-long training (short-term effect) and 4 weeks later (long-term effect). In addition, a field test investigated the effects of the training on reactive agility.

**Results:** A significant TIME *vs* GROUP effect was observed for (1) simple motor time ($p = 0.020$, $\eta p^2 = 0.08$), with improvement in the stroboscopic group in the post-test and retention test ($p = 0.003$, d = 0.42 and $p = 0.027$, d = 0.35, respectively); (2) complex reaction speed ($p < 0.001$, $\eta p^2 = 0.22$), with a large post-test effect in the stroboscopic group ($p < 0.001$, d = 0.87) and a small effect in the non-stroboscopic group ($p = 0.010$, d = 0.31); (3) saccade dynamics ($p = 0.011$, $\eta p^2 = 0.09$), with *post-hoc* tests in the stroboscopic group not reaching significance ($p = 0.083$, d = 0.54); and (4) reactive agility ($p = 0.039$, $\eta p^2 = 0.07$), with a post-test improvement in the stroboscopic group ($p = 0.017$, d = 0.49). Neither sensory sensitivity nor simple reaction time was statistically significantly affected as a result of the training ($p > 0.05$). A significant TIME *vs* GENDER effect was observed for saccadic dynamics ($p = 0.003$, $\eta p^2 = 0.226$) and reactive agility ($p = 0.004$, $\eta p^2 = 0.213$), with stronger performance gains in the females.

**Conclusions:** There was a larger effectiveness from the 6-week volleyball-specific training in the stroboscopic group compared to the non-stroboscopic group. The stroboscopic training resulted in significant improvements on most measures

Corresponding author
Michał Zwierko,
michal.zwierko@awf.wroc.pl

(three of five) of visual and visuomotor function with more marked enhancement in visuomotor than in sensory processing. Also, the stroboscopic intervention improved reactive agility, with more pronounced performance gains for short-term compared to the long-term changes. Gender differences in response to the stroboscopic training are inconclusive, therefore our findings do not offer a clear consensus.

## INTRODUCTION

In the fast-paced scenario of a volleyball game, players need to rapidly process a considerable amount of information in order to make appropriate motor action responses. In this regard, perceptual-cognitive abilities seem to be a crucial aspect of skilled performance under time pressure. Previous studies have identified several key perceptual-cognitive functions important for skilled player performance. Specifically, compared to non-athletes or novices, experienced volleyball players have an advantage in eye movement dynamics (*Fortin-Guichard et al., 2020*; *Piras, Lobietti & Squatrito, 2010*), the ability to track multiple moving targets (*Zhang, Yan & Yangang, 2009*), visuomotor processing (*Piras, Lobietti & Squatrito, 2014*; *Zwierko et al., 2010*), and also pattern recall, anticipation, and decision making (*De Waelle et al., 2021*; *Fortin-Guichard et al., 2020*; *Piras, Lobietti & Squatrito, 2014*).

In open-skill sports, expert players have a superior ability to interact with dynamic environments in real-time (*Araújo et al., 2019*). Expert-novice differences in the ability to identify relevant stimuli and fast information processing have been identified extensively in team-based sports, including soccer (*Klatt & Smeeton, 2022*), handball (*Blecharz et al., 2022*), baseball (*Müller, Fadde & Harbaugh, 2017*; *Ranganathan & Carlton, 2007*), basketball (*England et al., 2019*; *Zwierko et al., 2018*), and volleyball (*Fortin-Guichard et al., 2020*; *Trecroci et al., 2021*). While it is clear that perceptual proficiency enables athletes to produce a faster and more accurate motor response, through enhanced information processing and better perception-action coupling (*Farrow & Abernethy, 2003*; *Mann et al., 2007*), there is still ongoing research on improving athlete's reactive ability for specific sport environments.

Recently, stroboscopic training has been indicated as an effective tool to enhance perceptual-cognitive functions and physical performance (*Appelbaum & Erickson, 2018*; *Hülsdünker, Gunasekara & Mierau, 2021a, 2021b*; *Wilkins & Appelbaum, 2020*; *Wilkins, Nelson & Tweddle, 2018*). In general, the idea of stroboscopic training is based on an exercise with intermittent visual stimuli that force a greater demand on the visuomotor processing, leading to better performance in normal vision conditions (*Wilkins & Appelbaum, 2020*). Scientific evidence has consistently reported beneficial effects of specific stroboscopic intervention in general perceptual-cognitive and motor skills, *e.g.*, information encoding in short-term memory (*Appelbaum et al., 2012*), central visual field motion sensitivity and anticipatory timing (*Appelbaum et al., 2011*), visuomotor reaction
time (*Hülsdünker, Gunasekara & Mierau, 2021b*), coincidence-anticipation performance (*Ballester et al., 2017*), hand-eye coordination (*Ellison et al., 2020*), catching performance (*Wilkins & Gray, 2015*), and postural control in dynamic balance tasks (*Lee et al., 2022*). While studies have provided valuable information about the role of stroboscopic training on generic adaptation in perceptual-cognitive and motor skill performance in controlled environments, little is known about its role in the complex fast-paced motor tasks that are more specific to open-skill sports. Here, we aim to investigate the impact of stroboscopic intervention on reactive agility, which has been identified as an essential component of performance in team sports (*Sheppard et al., 2006*).

Recent evidence (*Hülsdünker, Gunasekara & Mierau, 2021a*, *2021b*) has shown in young athletes that stroboscopic training cause improvements in visuomotor reaction time, which was largely associated with visual processing. Specifically, more than 60% of the reduction in reaction time was due to the adaptations in latency of cortical potentials in brain regions for visual processing. Based on this data, it seems that training-induced changes may provide greater efficiency in visual function after training with a stroboscopic protocol. Therefore, we aimed at examining previously unexplored visual parameters following the stroboscopic training, such as: saccadic dynamics considered as a specific mode of exploratory eye movements that are accompanied by a shift of attention to the selected object (*Kowler, 2011*), and visual sensitivity identified as a quantitative index of cortical arousal in relation to physical exercise (*Davranche & Pichon, 2005*; *Lambourne & Tomporowski, 2010*).

Also, to the authors' knowledge, there are limited studies about the use of *in-situ* stroboscopic intervention in volleyball training. One of the studies in this area was conducted by *Kroll et al. (2020)*, who demonstrated the effectiveness of stroboscopic training in enhancing jumping performance and its potential as an addition to plyometric training in volleyball. Given that the utilization of stroboscopic protocols in a sport-specific context is a significant advantage (*Carroll et al., 2021*), research in this field is necessary for a practical application in sport training.

Based on the reported research gaps, the current study aimed at investigating the effect of stroboscopic training on young volleyball players. The 6-week program was based on *in-situ* sport-specific tasks performed either with or without stroboscopic eyewear. In line with previous studies (*Appelbaum et al., 2011*; *Hülsdünker, Gunasekara & Mierau, 2021a*, *2021b*), it was hypothesized that training with the use of stroboscopic eyewear would be more effective in improving visual and visuomotor performance, and in on-field test results measuring reactive agility, in relation to the same training without the stroboscopic eyewear. The potential impact of stroboscopic training will be also analyzed in relation to gender.

## MATERIALS AND METHODS

### Participants

To determine the minimum sample size required for this study, a power analysis was conducted using G*Power 3.1 (Heinrich Heine Universität Düsseldorf, Düsseldorf, Germany) (*Faul et al., 2007*). The analysis was based on an effect size of 0.25, an alpha of

**Table 1 Descriptive (mean ± standard deviation) characteristics of the experimental sample.**

|  | Stroboscopic group (*n* =25) | Non-stroboscopic group (*n* = 25) | *p* |
|---|---|---|---|
| Age (years) | 16.4 ± 0.7 | 16.6 ± 0.5 | 0.254 |
| Female (n) | 12 | 12 | 0.777[#] |
| Male (n) | 13 | 13 | |
| Height (cm) | 180.2 ± 8.2 | 181.9 ± 8.1 | 0.470 |
| Weight (kg) | 74.3 ± 10.4 | 71.6 ± 8.9 | 0.340 |
| Sports experience (years) | 6.7 ± 1.1 | 6.6 ± 1.3 | 0.732 |
| The effective time duration of training intervention (min/week) | 45.0 ± 1.4 | 46.1 ± 2.0 | 0.152 |

**Note:**
The '*p*' column corresponded to the t-test, except for one instance marked with '#' where it represented the chi-square statistic with Yates correction.

0.05, and a power of 0.95 for a mixed-model ANOVA with the between-subject factor of group (stroboscopic, non-stroboscopic) and the within-subject factor of time (pre, post, retention). The power analysis indicated that a minimum of 44 participants were necessary for the desired statistical power, which is consistent with the sample size used in a similar study by *Hülsdünker, Gunasekara & Mierau (2021a)*. Initially, 58 participants were recruited for this study. Due to random factors such as injury or illness, ultimately 50 athletes participated in the study (26 males and 24 females, age range 16–18 years, mean age ± SD 16.5 ± 0.6 years). The inclusion criteria for the study were: (a) volleyball training on a regular basis, at least 5 days a week, and (b) participating in official volleyball federation competitions during the season. The exclusion criteria included health conditions, such as epilepsy, migraine, or injury that prevented completion of the tests. For each gender, a random selection was made into either the stroboscopic group or the non-stroboscopic group (see Table 1 for a description of the experimental sample). All the participants, as well as their parent(s) or legal guardian(s), were informed of and consented to the testing procedures, and written informed consent was obtained for the use of the data from this research. The Research Ethics Committee of the University School of Physical Education in Wrocław reviewed and authorized the designed research protocol (No. 8/2021).

## Measurements

### Laboratory testing

To evaluate perceptual-cognitive functions (simple reaction speed, complex reaction speed, and sensory sensitivity), a computer-assisted Vienna Test System application (Schuhfried, Austria) was used. To this purpose, a computer (CPU 1.6 GHz) with a monitor (Dell P1913, diagonal 19″, resolution 1,280 × 1,024 pixels, refresh rate 85 Hz) and SCHUHFRIED response panel with foot-operated keys were used. Vienna Tests System is a reliable and valid psychometric tool used previously in sports vision training intervention (*Krzepota et al., 2015*).

### Simple reaction speed

To assess simple reaction speed, the S1 version of the Reaction Time test was administered. Participants were required to respond to randomly generated light stimuli in the form of a

yellow circle appearing at different time intervals (ranging from 2.5 to 6.0 s) at the bottom center of the screen. A total of 28 stimuli were presented during the test. To begin the test, participants placed their index finger on the 'waiting key' and in response to the stimulus, the finger was moved from the 'waiting key' to the 'response key'. After the response was made, the finger returned to its initial position. Participants were instructed to respond to the visual stimuli as quickly as possible. Two variables were calculated: simple reaction time (ms), defined as the interval of time between the appearance of the stimulus and lifting the finger from the 'waiting key', and simple motor time (ms), which characterizes the interval of time from lifting the 'waiting key' to pressing the 'response key'.

### Complex reaction speed

To assess complex reaction speed, the Determination Test version S1 was used. The Determination Test is a comprehensive and multifaceted reaction test that includes the presentation of both visual stimuli, such as colored stimuli and auditory signals, and is used in sports diagnostics (Ong, 2015). The test is designed to measure the ability to tolerate reactive stress, attention and complex reaction speed. Participants were to react to programmed randomly generated visual stimuli (in red, yellow, white, blue, and green colors) and auditory stimuli (high-pitch and low-pitch sounds) by pressing the appropriate button for the stimulus on the reaction panel. Also, white lights which appeared on a black background on the screen required pressing one of the two reaction pedals with a foot. The visual stimuli randomly appear in different locations on the screen. The rate at which the stimuli are presented was based on the participant's pace of work. The exercise length was approximately 6 min, including the practice phase. An analysis of the uncorrected total complex reaction speed (ms) was conducted. Additionally, incorrect responses in the complex task were monitored.

### Sensory sensitivity

The flicker-fusion frequency test is a widely used and objective measure of central nervous system function capacity, including cortical arousal and visual sensory sensitivity threshold (Davranche & Pichon, 2005; Fuentes et al., 2018; Mankowska et al., 2021). In this study the Flicker Frequency test was used to calculate the sensory sensitivity threshold. The test was performed under stable conditions by placing a light source in a darkened tube with backlighting background and held up to the participant's eyes. Inside the tube, a red point was placed in the center of vision. Initially, the participant recognized the light as steady (non-flickering), then the frequency of the red point gradually decreased until the subjective sensation of flickering. The participant should recognize the frequency change of the light source (steady to flickering) by pressing the button. The mean value of the measurements taken during the decreasing mode corresponded to the frequency (Hz) at which the light appeared to transition from a constant to a flickering state, as perceived subjectively. The decision to use only the descending series, instead of alternating descending and ascending, which is a more common method to avoid anticipation, was made due to time constraints related to the study procedure (Balestra et al., 2018).

### Saccade dynamics

To initiate a saccadic response, a free-viewing visual search activity was conducted, without any design specific to sports, where participants were instructed to identify a target (a red letter 'E') from a field of 47 distractors, including an inverted red letter 'E' ('ᴲ'), a red letter 'F', and a blue letter 'E' (*Zwierko et al., 2019*). Visual stimuli were displayed on a 55-inch television monitor (55UK6470; LG, Seoul, South Korea) at a distance of 1 m. Participants completed a total of 16 trials, with half of them being target-present trials and the other half being target-absent trials, and which were displayed in a random order. Participants were not informed about the type of trial they were in or the location of the target, ensuring an unbiased outcome. Participants were instructed to use one button to confirm the detection of the target (target-present trials) and another button to indicate the absence of the target (target-absent trials) as quickly as possible. During the visual search task, the saccadic movements of the eyes were recorded and analyzed. The average saccade velocity (°/s) was used to evaluate the saccade dynamics. Binocular gaze data was collected at a rate of 60 Hz (ETG 2w; SMI, Hamburg, Germany) using a portable eye-tracking system while participants performed the visual search task. A standard one-point calibration procedure was conducted binocularly, and the gaze data was subsequently analyzed using SMI BeGaze 3.5.101 software.

## Field test

### Reactive agility

Following the testing procedure by *Popowczak et al. (2020)*, a five-repetition shuttle run to test gates was conducted using a Fusion Smart Speed System (Fusion Sport, Coopers Plains, QLD, Australia). The system included an RFID reader to identify the athlete's tag, electronic gates with a photocell, an infrared transmitter, and light reflector, a smart jump mat integrated with a photocell, and computer software. The timing of movement during repeated stop-and-go directional adjustments in response to a random light signal at the gate was measured. Each participant performed five runs from the beginning mat to each of the designated gates (each 1 m long and positioned between photocells with reflectors). The distance from the mat to the gate was 4.5 m. Prior to beginning the measurements, the participants underwent a standardized warm-up procedure of 15 min. Each participant performed the test twice with a 3-min rest period interval. The shortest run time (s) was used for the analysis. Fig. 1 presents a schematic of the reactive agility test.

### Experimental procedure

The experiment involved a 6-week training period. The experiment was performed three times per week during the initial part of regular trainings. The pre-tests, post-tests and 4-week retention tests included both a laboratory test (perceptual-cognitive function) and a field test (reactive agility). The total duration of the testing session for one person did not exceed 60 min, with the experimental time for the laboratory sub-tests approximately 40 min, and for the field test 20 min (including warm-up). To minimize potential bias and ensure that the observed effects were truly due to the stroboscopic protocol, and also for logistical reasons, the participants were divided into four groups: a female stroboscopic

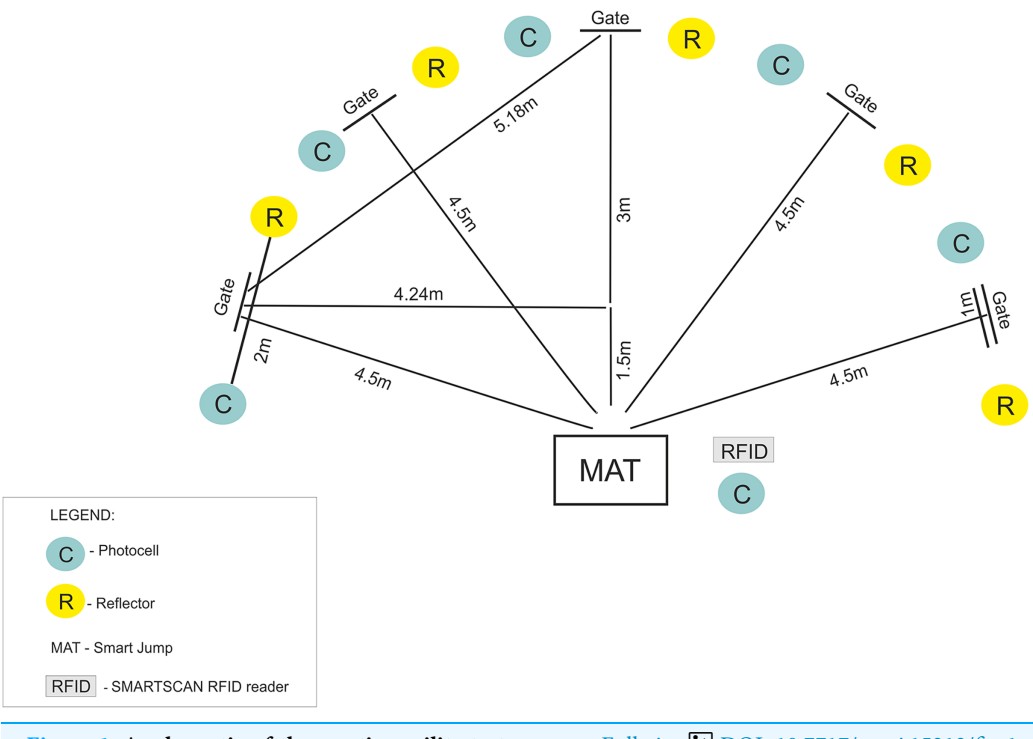

**Figure 1 A schematic of the reactive agility test.**

group, a male stroboscopic group, a female non-stroboscopic group, and a male non-stroboscopic group. Each group was tested on a separate day. Each testing session took place under the same conditions, starting at 9 am. The participants were familiarized with the tests (lab and field) prior to data acquisition using familiarization sessions to avoid any potential learning effects. Furthermore, a standard procedure, including an instruction and practice phase, was carried out in the Vienna Test System prior to each measurement. A method of randomizing the order of tests was used to control for potential order effects. Both, the stroboscopic and non-stroboscopic groups completed identical sport-specific exercises, under either stroboscopic or normal visual conditions, respectively. Each group trained on a separate court. Three volleyball-specific training protocols were undertaken. Protocol I 'wall passing drills' consisted of three tasks with the ball, including forms of reaction time exercise (*e.g.*, visual search, time pressure, second balls). Protocol II 'partner passing drills' consisted of five tasks in the frontal position, including forms of reaction time exercise by using external light stimuli, tennis balls, or time pressure. Protocol III 'passing rotation drill' consisted of two forms of passing (overhead and forearm passes) with changes of direction by forced time pressure. Fig. 2 presents a graphical illustration of the study protocols, along with examples of the exercises. Protocols lasted 25 and 30 min. During the training period, the experimental group wore stroboscopic eyewear (Senaptec Strobe, Beaverton, USA) with both lenses strobing. The eyewear was programmed and controlled *via* bluetooth using the Senaptec Strobe App installed on a smartphone. Following *Hülsdünker, Gunasekara & Mierau (2021a)*, stroboscopic protocols were restricted to 2.5 min with a 2.5-min break interval. The flicker speed of the stroboscopic

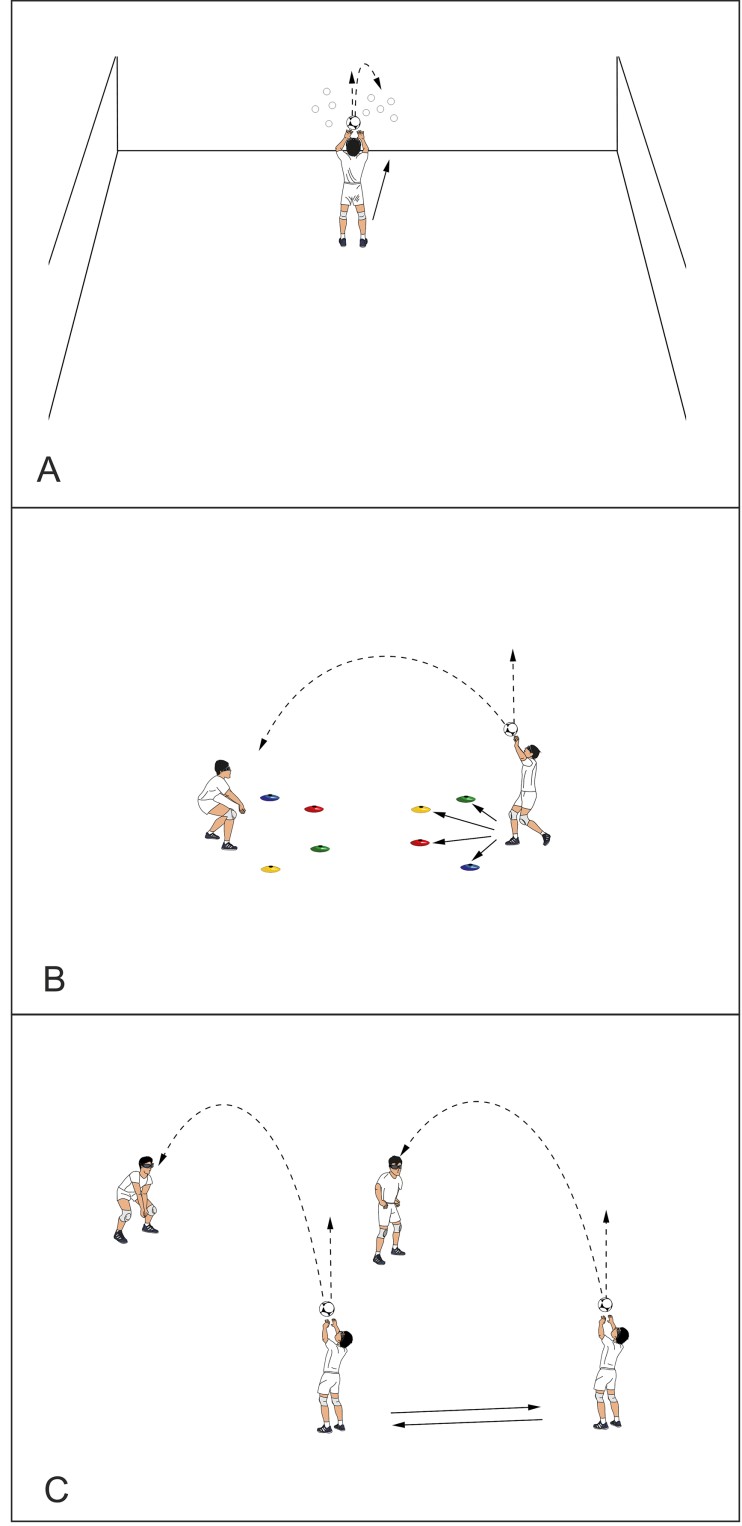

**Figure 2 A graphical illustration of the study protocols with examples of the exercises.** (A) Protocol I: "Wall Passing Drills": The player stands in front of a wall on which numbers are placed as optotypes. They bounce the ball off the wall, make an overhead pass above themselves, and then touch the designated number before returning to the starting position for the next bounce. (B) Protocol II: "Partner Passing Drills": The player makes an overhead shot above themselves, and on the second touch, passes

**Figure 2** (continued)
the ball to their partner. During the first pass, the partner signals one of four colors (blue, green, red, yellow). After the pass, the player must touch the designated color marker and return to the defensive position, and be ready for the next pass. (C) Protocol III: "Passing Rotation Drill": The players are arranged in two pairs and positioned facing each other. They make overhead or forearm passes to their partner, then simultaneously switch positions according to a rotation scheme to keep the ball in play. The graphic illustrations were prepared using the Easy Sports-Graphics software.

glasses was modulated for frequency (Hz) and duty cycle (%) (proportion of time the strobe light is on compared to the time it is off) to avoid adaptation effects, with the task difficulty increasing gradually (week 1: 15 Hz, 50%; week 2: 13 Hz, 50%; week 3: 11 Hz, 50%; week 4: 10 Hz, 50%; week 5: 9 Hz, 60%; week 6: 9 Hz, 70%).

### Statistical analyses

Descriptive statistics were presented as means and standard deviations. Normality of the data was examined using a Shapiro-Wilk test, and the homogeneity of variances was confirmed using a Levene test ($p > 0.05$). A mixed model ANOVA was performed, with inter-subject factor GROUP (stroboscopic, non-stroboscopic) and the intra-subject factor TIME (pre, post, retention). To investigate gender differences in response to the stroboscopic training, another mixed-model ANOVA was conducted with the inter-subject factor GROUP (female *vs* male) and intra-subject factor TIME (pre, post, retention). *Post-hoc* comparisons were adjusted using the Holm-Bonferroni procedure, and statistical significance was set at $p < 0.05$. The magnitude of the differences (effect sizes) was reported using Cohen's d and partial eta squared ($\eta p^2$) for t- and F-tests, respectively. The criteria for interpreting the magnitude of the effect sizes were: small (0.2), medium (0.5), and large (0.8) for the Cohen d and small (0.01), medium (0.06), and large (0.14) for partial eta squared (*Cohen, 1988*). JASP statistical software (version 16.1) was used for all analyses.

## RESULTS

The control analyses revealed no differences between the groups in anthropometric measures, gender, average training time per week, and training exposure time (as shown in Table 1). Pre-test performance scores between the stroboscopic and non-stroboscopic groups met the criteria for comparability in the control analysis, as all *p*-values were not significant ($p > 0.05$) for the following measures: simple motor time $p = 0.761$, simple reaction time $p = 0.842$, complex reaction speed $p = 0.351$, sensory sensitivity $p = 0.054$, saccade dynamics $p = 0.055$, reactive agility $p = 0.197$.

We used the mixed model ANOVA (main effects of TIME and GROUP) to analyze the variability of the perceptual-motor test results of the groups using stroboscopic glasses and without. The descriptive statistics of the sample in pre, post, and retention conditions for stroboscopic and non-stroboscopic groups are presented in Table 2. The interactions between the two-factor variables (TIME × GROUP) are displayed in Fig. 3.
**Table 2 Descriptive statistics of the visual, visuomotor and reactive agility parameters in the stroboscopic and non-stroboscopic groups in pre-tests, post-tests, and retention tests.**

| Variable | Group | Pre-test mean ± SD (min-max) | Post-test mean ± SD (min-max) | Retention-test mean ± SD (min-max) |
|---|---|---|---|---|
| Simple motor time (ms) | Stroboscopic | 133.40 ± 36.75 (80.00–222.00) | 118.80 ± 37.39 (61.00–219.00) | 121.20 ± 40.49 (77.00–227.00) |
| | Non-stroboscopic | 130.32 ± 34.51 (75.00–207.00) | 129.12 ± 32.18 (79.00–212.00) | 131.24 ± 26.61 (87.00–195.00) |
| Simple reaction time (ms) | Stroboscopic | 270.96 ± 44.51 (195.00–386.00) | 263.20 ± 35.98 (206.00–322.00) | 272.44 ± 41.03 (206.00–392.00) |
| | Non-stroboscopic | 273.72 ± 52.66 (220.00–447.00) | 272.56 ± 43.59 (206.00–393.00) | 273.64 ± 42.56 (203.00–401.00) |
| Complex reaction speed (ms) | Stroboscopic | 737.20 ± 62.49 (620.00–850.00) | 674.40 ± 61.58 (570.00–800.00) | 662.80 ± 81.27 (540.00–870.00) |
| | Non-stroboscopic | 719.60 ± 69.49 (630.00–870.00) | 697.20 ± 77.81 (600.00–860.00) | 684.80 ± 78.91 (580.00–890.00) |
| Sensory sensitivity (Hz) | Stroboscopic | 44.01 ± 5.69 (33.90–57.25) | 46.60 ± 5.22 (39.04–59.88) | 44.80 ± 6.23 (30.64–56.20) |
| | Non-stroboscopic | 46.84 ± 4.31 (39.91–55.36) | 46.68 ± 5.51 (38.53–58.83) | 47.08 ± 6.46 (37.51–59.88) |
| Saccade dynamics (°/s) | Stroboscopic | 94.30 ± 9.73 (81.03–116.93) | 99.26 ± 9.09 (84.17–119.07) | 101.38 ± 6.89 (88.44–116.58) |
| | Non-stroboscopic | 100.10 ± 11.07 (81.92–121.49) | 98.20 ± 9.37 (84.32–119.23) | 100.93 ± 8.51 (85.50–119.65) |
| Reactive agility (s) | Stroboscopic | 18.18 ± 1.23 (16.35–20.58) | 17.62 ± 0.98 (16.40–19.25) | 17.88 ± 1.32 (15.92–20.32) |
| | Non-stroboscopic | 18.61 ± 1.09 (16.24–20.40) | 18.39 ± 1.14 (16.00–20.35) | 18.04 ± 1.15 (16.19–20.80) |

### Simple reaction speed

The ANOVA on the motor time component of the simple reaction speed test revealed a significant main effect for TIME ($F_{2,96} = 4.59$, $p = 0.013$, $\eta p^2 = 0.09$) and no effect for GROUP ($F_{1,48} = 0.38$, $p = 0.542$, $\eta p^2 = 0.01$). Further, an interaction between the factors TIME and GROUP ($F_{2,96} = 4.06$, $p = 0.020$, $\eta p^2 = 0.08$) was observed. In the stroboscopic group, *post-hoc* tests showed significant differences between the pre-test and post-test ($133.40 ± 36.75$ *vs* $118.80 ± 37.39$ ms, $p = 0.003$, d = 0.42) as well as between the pre-test and retention test ($133.40 ± 36.75$ ms *vs* $121.20 ± 40.49$ ms, $p = 0.027$, d = 0.35), indicating a significantly faster motor time dependent on the stroboscopic training. The reaction time showed no significant main effect of TIME ($F_{2,96} = 1.61$, $p = 0.206$, $\eta p^2 = 0.03$), nor of GROUP ($F_{1,48} = 0.14$, $p = 0.709$, $\eta p^2 < 0.01$) nor interaction terms TIME × GROUP ($F_{2,96} = 0.96$, $p = 0.386$, $\eta p^2 = 0.02$).

### Complex reaction speed

Regarding complex reaction speed, we observed a statistically significant effect of TIME ($F_{2,96} = 81.54$, $p < 0.001$, $\eta p^2 = 0.63$) and the interaction TIME × GROUP ($F_{2,96} = 13.21$, $p < 0.001$, $\eta p^2 = 0.22$), however, the inter-subject effect—GROUP—was insignificant

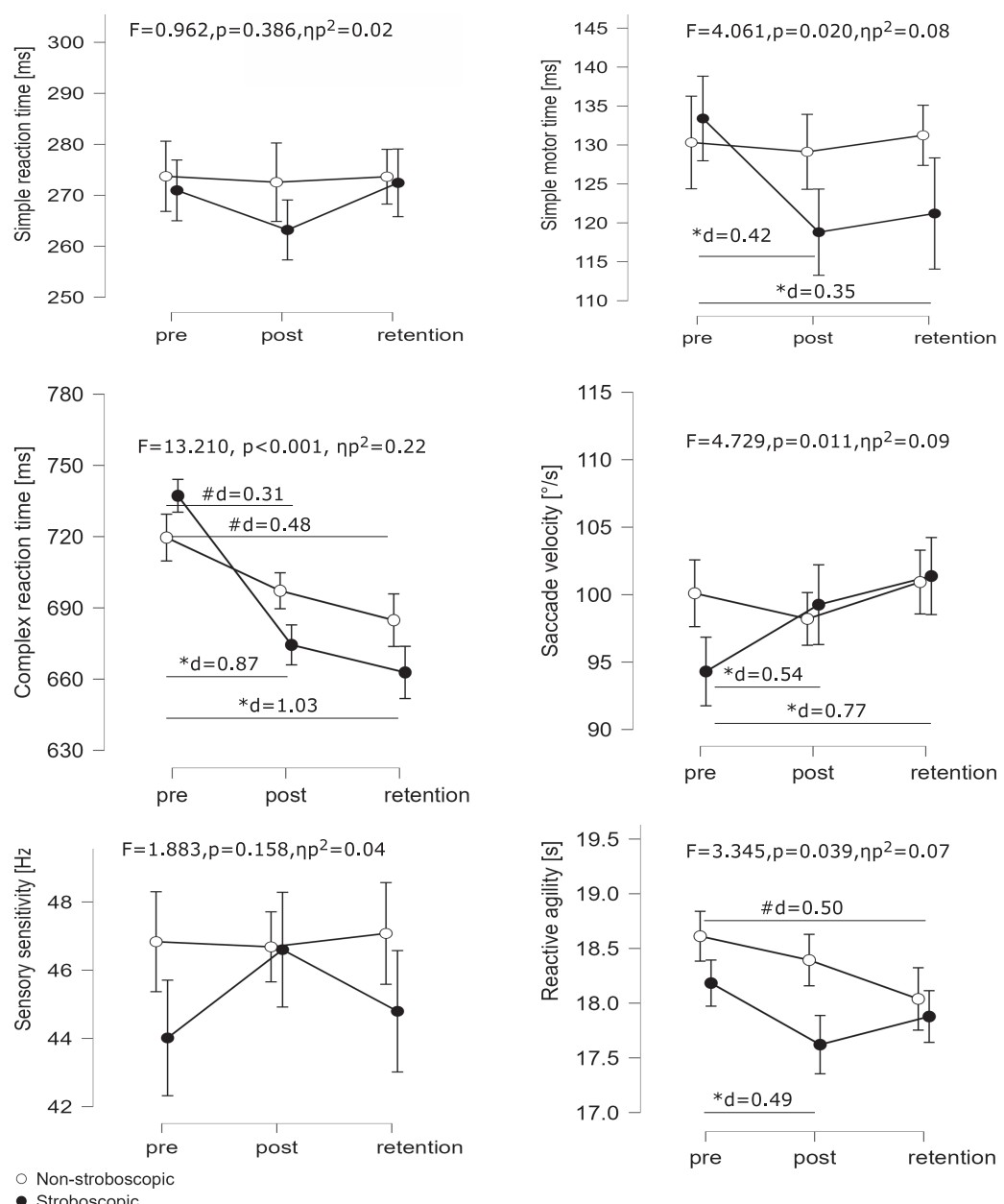

**Figure 3** **Interaction plots of visual and visuomotor parameters by TIME and GROUP: stroboscopic (black dots) *vs* non-stroboscopic (white dots) groups in a pre-post-retention design.** Pre-test, post-test, and retention test values are presented as means and 95% CIs. Significant changes ($p < 0.05$) in visual and visuomotor parameters are denoted by an asterisk (*) for the stroboscopic group and a pound sign (#) for the non-stroboscopic group, with accompanying effect sizes (d).

($F_{1,48} = 0.21$, $p = 0.649$, $\eta p^2 < 0.01$). *Post-hoc* analysis showed that both groups, stroboscopic and non-stroboscopic, achieved better results ($p < 0.05$) in the post-test and retention test compared to the pre-test, and the stroboscopic training yielded a higher magnitude of effect for the compared tests. In particular, pre-test *vs* post-test differences for the stroboscopic group showed a large effect (737.20 ± 62.48 ms *vs* 674.40 ± 61.58 ms,

$p < 0.001$, d = 0.87) and small effect for the non-stroboscopic group (719.60 ± 69.49 ms *vs* 697.20 ± 77.81 ms, $p = 0.010$, d = 0.31). Similarly, the pre-test *vs* retention test differences with a large effect were observed in the stroboscopic group (737.20 ± 62.49 ms *vs* 662.80 ± 81.27 ms, $p < 0.001$, d = 1.03) and differences with small effect for the non-stroboscopic group (719.60 ± 69.49 ms *vs* 684.80 ± 78.91 ms, $p < 0.001$, d = 0.48). The control analysis of complex reaction incorrect responses showed no significant main effect for TIME ($F_{2,96} = 0.91$, $p = 0.405$, $\eta p^2 = 0.02$) or GROUP ($F_{1,48} = 1.99$, $p = 0.164$, $\eta p^2 = 0.04$) factors. The interaction between the factors also had no significant effect ($F_{2,96} = 1.31$, $p = 0.276$, $\eta p^2 = 0.03$).

### Sensory sensitivity
The analyses of sensory sensitivity (flicker frequency) did not reveal a significant main effect either for TIME ($F_{2,96} = 1.34$, $p = 0.267$, $\eta p^2 = 0.03$) or for GROUP ($F_{1,48} = 1.68$, $p = 0.201$, $\eta p^2 = 0.03$) factors. The interaction between factors did not exert significant effects ($F_{2,96} = 1.88$, $p = 0.158$, $\eta p^2 = 0.04$).

### Saccade dynamics
Regarding saccade dynamics, the analyses revealed a significant effect of TIME ($F_{2,96} = 5.23$, $p = 0.007$, $\eta p^2 = 0.098$) and the interaction TIME × GROUP ($F_{2,96} = 4.73$, $p = 0.011$, $\eta p^2 = 0.09$). The factor GROUP did not reach statistical significance ($F_{1,48} = 0.43$, $p = 0.514$, $\eta p^2 = 0.01$). *Post-hoc* analysis in the stroboscopic group showed a trend toward significance when comparing the pre-test *vs* post-test results (94.30 ± 9.73°/s *vs* 99.25 ± 9.08°/s, $p = 0.083$, d = 0.54) and significance when comparing pre-test *vs* retention test results (94.30 ± 9.73°/s *vs* 101.38 ± 6.89°/s, $p = 0.002$, d = 0.77).

### Reactive agility
The analyses of reactive agility revealed a significant main effect of TIME ($F_{2,96} = 8.24$, $p < 0.001$, $\eta p^2 = 0.15$) and the interaction TIME × GROUP ($F_{2,96} = 3.35$, $p = 0.039$, $\eta p^2 = 0.07$). The factor GROUP was not significant ($F_{1,48} = 2.41$, $p = 0.127$, $\eta p^2 = 0.05$). In the stroboscopic group *post hoc* tests showed a statistically significant difference between the pre-test and post-test results (18.18 ± 1.23 s *vs* 17.62 ± 0.88 s, $p = 0.017$, d = 0.49), whereas in the non-stroboscopic group a significant difference between the pre-test and retention test results was observed (18.61 ± 1.09 s *vs* 18.04 ± 1.15 s, $p = 0.014$, d = 0.50).

### Gender differences in response to stroboscopic training
Table 3 presents the differences in the test results between male and female players within the stroboscopic group over time of the experimental procedure. A significant TIME *vs* GENDER effect was observed for saccadic dynamics ($F_{2,46} = 6.72$, $p = 0.003$, $\eta p^2 = 0.23$) and reactive agility ($F_{2,46} = 6.21$, $p = 0.004$, $\eta p^2 = 0.21$). *Post-hoc* analysis indicated that in both cases, a significant improvement in post-test results compared to pre-test results occurred in the female group (95.65 ± 8.61°/s *vs* 87.08 ± 5.18 °/s, $p = 0.018$, d = 1.14 for saccadic dynamics and 18.26 ± 0.64 s *vs* 19.23 s, $p < 0.001$, d = 1.46 for reactive agility,

**Table 3** Results of visual, visuomotor and reactive agility tests in relation to gender within the stroboscopic group: descriptive and statistical values in pre-tests, post-tests, retention tests, and interaction effects (TIME × GENDER).

| Variable | Group | Pre-test mean ± SD | Post-test mean ± SD | Retention-test mean ± SD | F | p | ηp² |
|---|---|---|---|---|---|---|---|
| Simple motor time (ms) | F | 157.92 ± 35.54 | 146.09 ± 33.77 | 146.42 ± 41.78 | 0.21 | 0.809 | 0.01 |
| | M | 110.77 ± 19.70[bb] | 93.62 ± 17.87[abb] | 97.92 ± 21.36[bb] | | | |
| Simple reaction time (ms) | F | 289.25 ± 54.87 | 280.33 ± 40.41 | 288.42 ± 50.31 | 0.13 | 0.875 | 0.01 |
| | M | 254.08 ± 23.67[b] | 247.39 ± 23.00[b] | 257.69 ± 23.58[b] | | | |
| Complex reaction speed (ms) | F | 755.83 ± 56.96 | 690.83 ± 59.92[aa] | 691.67 ± 75.30[aa] | 2.29 | 0.113 | 0.09 |
| | M | 720.00 ± 64.55[bb] | 659.23 ± 61.44[aabb] | 636.15 ± 80.06[aabb] | | | |
| Sensory sensitivity (Hz) | F | 44.07 ± 6.51 | 47.63 ± 5.82 | 43.76 ± 4.80 | 1.44 | 0.247 | 0.06 |
| | M | 43.96 ± 5.08 | 45.66 ± 4.63 | 45.75 ± 6.22 | | | |
| Saccade dynamics (°/s) | F | 87.08 ± 5.18 | 95.65 ± 8.61[a] | 100.70 ± 7.43[aa] | 6.72 | 0.003 | 0.23 |
| | M | 100.97 ± 8.04[bb] | 102.59 ± 8.49 | 102.00 ± 6.59 | | | |
| Reactive agility (s) | F | 19.23 ± 0.79 | 18.26 ± 0.64[aa] | 19.03 ± 0.66 | 6.21 | 0.004 | 0.21 |
| | M | 17.21 ± 0.56[bb] | 17.03 ± 0.62[bb] | 16.81 ± 0.72[bb] | | | |

**Notes:**
[a] Denotes significant differences within groups (between post-test *vs* pre-test and retention-test *vs* pre-test).
[b] Denotes significant group difference (female *vs* male).
[a/b]$p < 0.05$.
[aa/bb]$p < 0.01$.
F-female, M-male; F-interaction effect (TIME × GENDER).

respectively), with no significant changes in the male group ($p > 0.05$). No significant gender differences ($p > 0.05$) were found in the course of variability of other test results.

## DISCUSSION

Our study evaluated the effects of stroboscopic intervention on visual, visuomotor and reactive agility in youth volleyball players. The results indicated that stroboscopic training were more effective than regular training, with the stroboscopic group showing significant short and long-term improvements in simple motor reaction time and saccade velocity, as well as larger gains in complex reaction speed and reactive agility (in the short-term). Additionally, gender differences in response to the stroboscopic intervention were identified in two out of six variables.

Our findings add further support to the suggestions (*Carroll et al., 2021*; *Wilkins & Appelbaum, 2020*) that stroboscopic training may be an effective tool for improving visual and visuomotor abilities in sports training. Previous behaviors and neurophysiological evidence (*Hülsdünker, Gunasekara & Mierau, 2021a*, *2021b*) showed that visuomotor reaction times after stroboscopic protocols in young elite badminton athletes were largely associated with visual processing rather than motor processes. Similarly, *Poltavski, Biberdorf & Praus Poltavski (2021)* used electrophysiological indexes of EEG and visual evoked potentials to monitor visual training effects and progress in the performance of youth ice hockey players. Training-induced changes in the electrophysiological indexes reflected greater efficiency in visual information processing and cognitive resource allocation following visual training with a stroboscopic protocol. However, the present study results cannot fully confirm the cited data. The laboratory parameters analyzed in

our study, *i.e.*, a visual sensory sensitivity measured by flicker frequency and simple reaction time, did not show significant variations after the reactive training in either the stroboscopic or non-stroboscopic group. On one hand, volleyball is a very visuomotor-demanding sport and may yield beneficial neuroadaptation to visual sensory processing in earlier stages of sports training. In a study using electrophysiological recordings (visual evoked potentials, VEP) in young athletes practicing volleyball for 2 years, *Zwierko et al. (2014)* observed a decrease in VEP latencies (P100 and N75), suggesting that the effects of regular sports training cause improvements in the sensory stage of information processing. On the other hand, it is also possible that the maturation of the visual system may influence training effects in different ways between athletes aged 13 (*Hülsdünker, Gunasekara & Mierau, 2021b*) and 16–18 (current study). Following *Kovács et al. (1999)* the maturation of the visual system is not homogeneous, and the development in visual spatial integration may extend between 5 and 14 years. That is why stroboscopic training may bring more noticeable effects in younger athletes. It is also possible that the sensory processing measurement used in this study (reaction time to a yellow light appearing on a computer screen, flashing light in a sensory sensitivity test) primarily activated processing in the ventral visual stream, which is responsible for processing visual information such as object recognition and color perception (*Ungerleider & Mishkin, 1982*), while the environment of the volleyball players mainly requires the use of the dorsal visual stream, which processes more coarse visual information such as motion, depth and spatial location. The results indicate that athletes required to quickly respond to moving stimuli rely more on their dorsal stream (*Sasada et al., 2015*). However, research suggests that there is an interaction between the dorsal and ventral pathways in action contexts (*van Polanen & Davare, 2015*), so stroboscopic excitation of the dorsal pathway may influence the ventral pathway. It is important to note that this is just a possible explanation and that further research would be needed to determine the specific mechanisms behind the observed effects.

It appears that stroboscopic protocols may have specific effects on certain types of tasks, as demonstrated in our study, where it improved simple motor time (the speed of a simple reaction involving hand movement) in the stroboscopic group but not in the non-stroboscopic group. The adaptation to stroboscopic training used in sport-specific exercises, by better processing and integrating visual information for efficient movement execution, led to improved coordination and an increase in movement speed. This aligns with previous studies, which have found that stroboscopic protocols are effective in enhancing eye-hand coordination in dynamic and complex coordination tasks (*Ellison et al., 2020*; *Jones, Carnegie & Ellison, 2016*).

This study gives partly supporting evidence to the claim that stroboscopic protocols improve the speed of cognitive and information processes created by the action of multiple stimuli on the player sensory system. We observed a significant effect of the 'TIME' factor in the complex reaction speed, which suggests that both groups improved their reaction speed in complex tasks, with more pronounced gains in the stroboscopic group. The main focus of the protocols (especially protocol 2) was to perform specific volleyball exercises that involved complex reaction tasks (*e.g.*, as shown in the Fig. 2B). It is also possible that

the results of the complex reaction tests may have been influenced by a learning effect, despite the participants having prior familiarity with the tests (both in the lab and field). This is an issue that has been observed in previous studies using a stroboscopic protocol (*Wilkins, Nelson & Tweddle, 2018*). A number of previous studies have shown that the stroboscopic training leads to improvements in some aspects of visual cognition, in particular central visual field motion sensitivity and transient attention abilities (*Appelbaum et al., 2011*), some aspects of visual memory (*Appelbaum et al., 2012*), anticipatory timing (*Ballester et al., 2017*; *Smith & Mitroff, 2012*) and Go/No-Go reaction time (*Appelbaum et al., 2016*). There is also some evidence which does not confirm the positive effect of stroboscopic training on visual and perceptual-cognitive skills in elite athletes (*Wilkins, Nelson & Tweddle, 2018*). It is possible that these conflicting results may be caused by differences in sporting experience and individual adaptability to stroboscopic conditions.

To our best knowledge, this is the first study where the positive impact of stroboscopic training on saccadic dynamics was reported. In stroboscopic conditions, a player is forced to utilize the limited visual samples, which may cause increased efficiency in the oculomotor system. In consequence, the temporal integration of information during stroboscopic protocols becomes more efficient (*Wilkins & Appelbaum, 2020*), leading to a saccadic dynamics advantage later. This finding corresponds to the study results by *Poltavski, Biberdorf & Praus Poltavski (2021)*, who observed that training of information processing skills based on visuomotor integration and information processing skills (visual software training) in ice hockey players, indirectly trained their oculomotor system (eye quickness variables, *i.e.*, near-far quickness and target capture tasks). It should be noted that in the current study the baseline level between the stroboscopic and non-stroboscopic groups was close to achieving significance ($p = 0.055$), which may to some extent affect the performance gains in the stroboscopic group being attributed to a lower starting level. The issue of oculomotor processes when performing tasks with a repeated interruption of visual input is worth further investigation. Future research should try to evaluate the eye movements that lead to the successful tracking of moving objects in stroboscopic conditions.

Reactive agility improvements in the post (stroboscopic group) and retention test (non-stroboscopic group) were further observed in the field test; however, final training effects did not differ between groups. While both groups showed improvement in reactive agility, the study results indicate a clear positive impact of stroboscopic intervention on reactive agility. However, it is important to note that reactive agility is also closely associated with motor and biomechanical components, such as running speed and technique, balance, strength and muscle power of the lower limbs (*Condello et al., 2016*; *Freitas et al., 2022*; *Koźlenia et al., 2020*; *Spiteri, Newton & Nimphius, 2015*; *Spiteri et al., 2014*). It appears that these observations support the idea that the gains seen in the present study, apart from the perceptual-cognitive factor, may be mainly linked to improvements in strength, power, or other factors, as similar findings have been observed in previous research on reactive agility in male and female volleyball players. In particular, our own previous studies *Zwierko et al. (2022)* have found that factors such as lower body explosive strength, complex reaction

time, selective attention, and sensory sensitivity play significant roles in determining reactive agility, with adjusted coefficients of determination ($R^2$) of 23.6% (explosive strength and complex reaction time) and 34.5% (explosive strength, selective attention, and sensory sensitivity). Furthermore, other studies *Horníková, Jeleň & Zemková (2021)* have found that perceptual-cognitive factors contribute to 23.6% of determining reactive agility in team sport players. The contribution of perceptual-cognitive factors in reactive agility seems to be significant.

We found significant gender differences in relation to test result variability only for two of the six analyzed parameters, *i.e.*, saccade dynamics and reactive agility, when the females received improvements with large effect sizes, while the males did not. It is possible that the stronger performance gains in the female group may be attributable to a lower baseline performance level (pre-test value in Table 3). Our previous study (*Zwierko et al., 2022*) on a cohort of 135 participants indicated that male volleyball players highly outperformed female volleyball players in reactive agility test and saccade dynamics, whereas, perceptual and cognitive factors (selective attention, simple reaction speed, complex reaction speed, sensory sensitivity) presented non-significant or small to moderate differences in relation to gender. *Shaqiri et al. (2018)* reported that the gender-related differences in visual and perceptual function are heterogeneous. Using fifteen different visual tasks, they found that males had significantly better performances than females in simple reaction time, visual acuity, visual backward masking, motion direction detection, biological motion, and the Ponzo illusion. In our view, the observed gender differences in response to stroboscopic intervention are not straightforward or easy to interpret, as there is no clear consensus in previous research. It appears that certain types of tasks are more impacted by stroboscopic training than others. The effects may be specific to the task being performed and may also vary based on gender. In a future study we will try to identify the best parameter or parameter combination to distinguish responders from non-responders in response to stroboscopic intervention, and gender will be considered an independent variable.

The present study provides novel insights into the relationship between stroboscopic training and visual and visuomotor performance in young volleyball players. However, this study is not free of limitations. First, in our study, there is a lack of specific volleyball skill tests that should be designed to transfer standardized laboratory tasks (near transfer) to more sport-specific conditions (far transfer). Achieving far transfer after different kinds of perceptual and cognitive training is so far poorly investigated in research studies (*Fleddermann, Heppe & Zentgraf, 2019*; *Palmer, Coutts & Fransen, 2022*; *Sala et al., 2019*). It would be interesting for a future study to evaluate the effects of stroboscopic training on specific sporting performance. Second, the experimental sample was limited to young athletes, and it has recently been reported that stroboscopic vision may be used to stimulate larger training effects in more skilled players (*Beavan et al., 2021*). Therefore, the generalizability of the present results to other athlete age categories needs to be tested in future studies. Third, the findings of this study should be evaluated in the context of the limitations in the study design. In traditional controlled designs, such as drug trials, it is possible to establish a placebo protocol to balance the experience, expectancy and motivation of the participants. However, in the case of stroboscopic protocols, it is not

possible to create a placebo or conceal the experience of the stroboscope from the participants (*Ellison et al., 2020*; *Wilkins & Appelbaum, 2020*). To minimize potential bias, we divided the participants into four separate groups and trained them on different courts. Additionally, the training protocols were conducted by one instructor to ensure consistency. Our study design did not include measures of motivation and expectancy, which can be considered a limitation of the study.

## CONCLUSION

The present results demonstrate a significantly greater effectiveness of 6-week volleyball-specific training in the group using stroboscopic eyewear compared to those who trained using regular methods. Stroboscopic intervention resulted in significant improvements in most measures (three of five) of visual and visuomotor function, with greater enhancement in visuomotor than in sensory processing. Additionally, the stroboscopic training improved reactive agility, with more pronounced performance gains in short-term changes compared to long-term changes. Gender differences in response to the stroboscopic training are inconclusive, and our findings do not offer a clear consensus on this issue.

### Funding
The authors received no funding for this work.

### Competing Interests
The authors declare that they have no competing interests.

### Author Contributions
- Michał Zwierko conceived and designed the experiments, performed the experiments, analyzed the data, prepared figures and/or tables, authored or reviewed drafts of the article, and approved the final draft.
- Wojciech Jedziniak conceived and designed the experiments, performed the experiments, analyzed the data, prepared figures and/or tables, authored or reviewed drafts of the article, and approved the final draft.
- Marek Popowczak conceived and designed the experiments, performed the experiments, prepared figures and/or tables, authored or reviewed drafts of the article, and approved the final draft.
- Andrzej Rokita conceived and designed the experiments, authored or reviewed drafts of the article, and approved the final draft.

### Data Availability
   The raw measurements are available in the Supplemental File.

## Supplemental Information

Supplemental information for this article can be found online at http://dx.doi.org/10.7717/peerj.15213#supplemental-information.

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
