# Peer review of "Effects of in-situ stroboscopic training on visual, visuomotor and reactive agility in youth volleyball players"

_PeerJ, doi:10.7717/peerj.15213_

## Round 0.1 · original submission · Major Revisions

Your manuscript entitled "Effects of reactive training in youth volleyball players: stroboscopic vs non-stroboscopic interventions" which you submitted to PeerJ, has now been reviewed by three independent experts. Their comments are included .

As you will see, the reviewers express some reservations about your manuscript and request that additional work be done to attempt to address these concerns. If you believe that your manuscript can be adequately revised to answer these points, I will be happy to consider a revised version, although it is likely that a re-review will be necessary before a final decision can be made on publication.

Reviewer 1 ·

Basic reporting

• Figure 1 is really nice but it’s quite cramped. Perhaps you could report the ANOVA results below each graph, which would then allow you to make each one wider and less congested. Also, you should be consistent in terms – here your y axis is titled flicker frequency (the test), whereas the other y-axes are titled as the component that they are measuring
• Table 2 is a bit of a mess – I would suggest sticking to 2 dp throughout, and also presenting this in landscape
• Line 331-332 – “leading to saccadic dynamics advantage later” is not what the Wilkins & Appelbaum reference refers to. You can use this citation for the “integration of information during stroboscopic protocols becomes more efficient” part, but not after this. This then creates the issue of how you are explaining the increase in saccadic dynamics….
• Line 41 – stated that simple reaction time not significantly affected, but on line 32 it is reported as being so. This distinction between the two measures of simple reaction time is not always clear and needs to be carefully worded throughout
• Line 27 – should specify these are volleyball specific/in-situ volleyball-based, or something along those lines
• The abstract background needs to explain what stroboscopic training is. Whilst the abstract is quite long, this is important. The writing could be more concise elsewhere to shorten it overall (in particular, the reporting of the results, e.g., “d” not “Cohen d”, “p” not “corrected p-value” – at least in the abstract section).
• 88-91 – is there a reason this, of all the studies mentioned, is reported in slightly more detail? Is there a particular relevance of this study to the present paper? If not, it might be worth removing. If so, then this relevance needs expanding on.
• Lines 286-295 would be better off in the introduction. Likewise 327-328
• Lines 266-267 – delete. This is the first time you’ve chosen to summarise one of the results sections – it is unnecessary
• Line 47-48 – not sure what is meant by this? Is this referring to the retention tests finding fewer significant results? Could be clearer
• 55-56 – “often to unexpected stimuli”…really?
• Line 204 – better to term stroboscopic vs non-stroboscopic than intervention vs control
• Line 236 – pre-test not initial state
• Numbers less than 10 should be written as text (e.g., four not 4)

Experimental design

• Why was the flicker frequency test chosen for sensory sensitivity? It seems to have no relevance to volleyball skills and therefore is not surprising that it produced non-sig results. But a difference test of “sensory sensitivity” (which is a fairly broad term that is never actually discussed or defined in the introduction) could have.
• Simple reaction speed – is the stimuli appearing in the same location each time? If so, where?
• Complex reaction speed – more detail is needed here, or reference to a paper that has more detail. Reference to a paper would also be useful for sensory sensitivity
• Saccade dynamics – were the targets and distracters stationary? Did participants know when they were in a target-present or target-absent trial? What exactly were the measure(s) taken from the gaze data? (I see it is eventually specified on line 279/280, but this needs proper detailing here)
• Reactive agility – how far away are the gates from the beginning mat?
• Protocol – the protocol descriptions are not the most clear. To me, it seems that these are standard volleyball drills often done during training. If this is the case, then the paper should be reframed with that in mind, rather than trying to claim it as reactive training. The idea that standard volleyball training with stroboscopic glasses can enhance various visual-perceptual skills to a greater extent than non-stroboscopic conditions has relevance, whereas the introduction of the terms “reactive training” is actually confusing matters.
• Protocol – is it necessary to state the four separate training groups? This confused me – what was its purpose? Just logistical? What you do need to explain, though, is whether the participants were aware of what others were doing…specifically, if the non-stroboscopic group were aware of the stroboscopic training group, then this is a significant limitation to the study (impacts motivation, expectation, etc)
• Protocol - need to explain what is meant by duty cycle in this section.
• Any gender differences? This would seem to be a simple analysis to do and would be interesting to see.

Validity of the findings

• Line 317 goes against the findings of your simple reaction measure (i.e., if this is the case, the reaction time measure should have had sig results)
• Line 343-344 – I don’t see how you can make this statement, given the retention test results (that the non-stroboscopic group actually saw sig. improvements whereas the stroboscopic group did not). The retention tests need to be acknowledged, with a potential explanation suggested.
• Lines 86-87 – reading this, it implies that your work is going to have a nice transfer test and measure of motor skills…but it doesn’t at all. I would delete this. Your work is contributing to the literature on how stroboscopic training impacts visual-perceptual skills (and a test of reactive agility), but it is not addressing transfer and/or motor skills

Additional comments

Thank you for the opportunity to review your work. This is a nice study design that, in my opinion, warrants consideration for publication pending some considerable revisions. Key to this are:

1), re-framing the paper more simply in terms of stroboscopic vs non-stroboscopic training, without the unnecessary inclusion of "reactive training", which I believe confuses things more than it helps matters,
2) the issue of motivational/expectancy differences between the two groups. More information is needed to address this.
3) the choice of test for sensory sensitivity is strange, given the activity (volleyball). I wouldn't expect sig. differences to emerge here, so this should be acknowledged, but in reality, it feels like a poor (but not fatal) design decision.

I would also encourage you to explore potential gender differences given the nice sample size and data collected, and also, throughout the paper the detail and explanations could be improved (see specific comments above).

·

Basic reporting

no comment
I don't feel qualified to judge about the English language and style.

Experimental design

no comment

Validity of the findings

no comment

Additional comments

The study is well designed according to the requirements of the scientific method and based on solid literature.
Your research is valuable in terms of its subject and scope. The topic is relevant, and the study can contribute to the extant literature.
The paper is generally well written and the conclusions are interesting and important for sports practice.
My suggestion - add study limitation, after discussion.

·

Basic reporting

- The article is (mostly) written in a clear and proficient English.
- Background is sufficient, although some further explanations are needed (e.g. in the introduction) and some arguments in the discussion need further clarification.
- Structure is in line with the PeerJ instructions.
- The plots in figure 1 are very small and likely hard to read in the printed version. There are too many decimal places in table 2 which should be changed (1 decimal place maximum).
- Some more literature is needed in the discussion section

Experimental design

- The experimental study using a pre-post-retention design provides valuable information to the field of stroboscopic training in sport.
- The research question is clear however, the research gap needs further elaboration to point out the contribution of this study to existing knowledge.
- Experimental approach is solid, although more details are needed on the experimental protocol and the participant characteristics.

Validity of the findings

- Data is reported with sufficient detail. Some changes have to be made to improve readability.
- Conclusions are mostly supported by the data, however some aspects in the discussion require a more in-depth analysis.

Additional comments

The authors investigated the short- and long-term effects of a stroboscopic training intervention on visual, visuomotor and reactive agility performance in volleyball players. Laboratory and field-based tests were performed prior to and after a 6-week training intervention and following another 4-week retention period. Results suggest a positive effect of stroboscopic eyewear especially on visuomotor functions which was interpreted as further evidence supporting the use of stroboscopic training intervention in volleyball.
The experimental study is well designed and conducted with a high number of participants which provides interesting insights into the effectiveness of stroboscopic training in sports.

4. General comments

Abstract

The terms ‘reactive ability’ and ‘reactive training’ are confusing. I assume that it refers to reaction time however it is also often used in the context of e.g. plyometric training and stretch-shortening-cycles. This should be clarified to avoid confusion.

l.27: The abstract is well written however, it should be indicated if the training tasks were sport-specific or generic. Since the authors highlight that the advantage of stroboscopic training is its use in sport-specific settings, this information should be included already in the abstract.

Introduction

l. 64: The authors state that ‘From a dynamic perspective…’. What is meant by a dynamic perspective? Does this refer to dynamic vision or the dynamics of the sports (e.g. ball and team sport) in general. Please add a few words what is meant by ‘dynamic perspective’.

ll.79: The possibility to use stroboscopic training in the context of sport-specific training is highlighted, however, further information is missing. Please provide some more details, why the use of stroboscopic training in sport-specific training settings is considered a strength.
This is of particular importance since the following paragraph mentions improvements following stroboscopic training that are generic rather than sport-specific. It is thus unclear why the sport-specific training is mentioned. I would recommend introducing the sport-specificity of stroboscopic training after the generic adaptations mentioned in lines 80-85. It also better links to the distance-transfer in line 86.

l. 86 I assume distance transfer refers to the classification of near, middle and far-transfer. Since there are different classification systems and terms for transfer, please clarify that when talking about distance transfer, this means to sport-specific abilities. In the limitations section the authors use near- and far-transfer which in my opinion would fit better also in the introduction.

Further it is not clear to me why the research gap especially applies to young athletes, while in the next sentence the authors mentioned that Hülsdünker et al. 2021 performed their study in young badminton players. Please clarify.
Overall, the research gap needs further elaboration. The authors mention missing distance-transfer analysis especially in younger athletes but the study they refer to (Hülsdünker et al. 2021) is described as investigating this. Do the authors assume that results would be different in volleyball due to the sport characteristics or that adaptations may be observed in other parameters that were not investigated by Hülsdünker et al. 2021?

L. 92 Based on the comment above, be more clear what the research gaps are and how the study contribute to the existing body of research.

Materials and methods

l. 104: Where is the effect size coming from (estimation or based on previous research) and what statistical model was used for the a-priori calculation (mixed-model ANOVA?). Please clarify.

l. 106: Since it is a longitudinal study, include information on the drop-out rate due to injury, illness, or other reasons.

l. 108: The average training hours per week should be added to table 1.

l. 142: What is the ‘total complex reaction speed’? Is this the raw reaction time derived from the Vienna Test system or a calculated value (e.g. correcting for error rate)? Please specify what variable has been used.

l. 208: In the statistics section you state that eta square (η²) is reported however, in the results you report partial eta square (ηp²). Please correct.

- Were the participants familiar with the performance tests (lab and field) or were there familiarization sessions included prior to the data acquisition to avoid learning effects? Especially the results on the complex reaction time may be influenced by getting familiar with the task.

- How did the authors control for potential order effects? Was the order of tasks randomized/counterbalanced or was the same test order used for each participant?

- Provide more information on the overall test duration as well as the experimental time for the different sub-tests (laboratory and field).
- An illustration of the performance assessment especially for the reactive agility test and the training intervention would help getting a better idea of the intervention. Otherwise, it is unclear to anyone who is not familiar with volleyball how e.g. ball-passing drills are associated with reaction time.

- Training time according to the intervention protocol should be added to table 1 to compare the exposure time to the training intervention between the groups.

- Control analyses should confirm no statistically significant difference between groups in terms of training experience, training load per week, training exposure time and pre-test performance scores to ensure comparability

Results

Overall, the results section provides a lot of detail which is however confusing at some points to the number of values that is reported in the text and some inconsistencies in the comparisons between the intervention and control group. This should be streamlined to improve readability (see detailed comments below).

- Reduce the decimal places for reported raw data. Due to the number of decimal places, the results are hard to read. One decimal place is sufficient. This applies to both the text and table 2.

- Units are missing (e.g. in line 225, 238, 239, …). Please check the results section.

- Instead of ‘Cohen d’ use the letter ‘d’ for post-hoc test effect sizes. It has already been defined in the statistics section. Same applies to the p-value. No need to define it as ‘corrected p’ since the correction method has been introduced in the statistics. Just state the ‘p’. Please also change this in the abstract.

ll. 255: What about the results in the control group on saccade dynamics. Since you have an interaction effect, the difference between the intervention and control group should be reported.

ll. 266: This is a results interpretation which belongs to the discussion section. Further, this conclusion is based on a comparison of pre-post for the intervention and pre-retention for the control group. It is unclear how this supports a positive impact of stroboscopic training. To draw this conclusion, the pre-post comparison of significant results in the intervention group and non-significant results in the control group, would suggest more pronounced performance gains in the stroboscopic training group (although only short-term).

Discussion

ll. 270-283: The result summary at the beginning of the discussion is too long and more a repetition of the results rather than a summary of the most important findings. This should be streamlined and unnecessary information (e.g. number of effects sizes) excluded from this part.

ll. 295-310: The authors compare their results to previous research by Hülsdünker t al. 2021 and Poltavski et al. 2021. It is argued that the conflicting findings may be attributable to the age of participants. However, it should also be discussed that the test on the Vienna Test system uses a flashing yellow circle which activates the ventral visual stream while volleyball players are exposed to an environment that mainly requires the dorsal visual stream. Since these two systems adapt differently in athletes, this could be another explanation for the missing effects on reaction time.

The authors should also provide an explanation for the observed effects in the simple motor time that improved ion the stroboscopic but not the control group.

l. 314: The term ‘greater results’ is misleading. I assume the authors mean stronger performance gains.

ll. 311-325: How do the authors explain the very strong effect of the factor ‘TIME’ in the complex reaction test indicating that both groups improved although the performance gains were more pronounced in the intervention group. Was this related to the general training content that was more focused on reaction and decision making when compared to the “regular” training or could it be explained by a learning effect over trials, if participants were unfamiliar with the determination test in the pre-test assessment?

ll. 326-335: The results on saccadic eye movement indicate a comparatively large baseline difference between the two groups (figure 1). This should be mentioned when discussing the results (including a control analysis on the baseline performance) especially since the two groups were not different in the post- and retention test. If there are differences at baseline, the performance gains in the intervention group may be attributable to a lower starting level.
ll. 338-339: Do the 23.6% correspond to the same test the authors used in their study? If yes, this would strengthen the argument, that performance gains may be related to improvements in strength, power or other factors.
However, then the conclusion is confusing stating that stroboscopic training is promising to improve reactive agility. At least, the authors should differentiate between short-term and long-term effects they observed for the reactive agility test.

---

## Round 0.2 · Minor Revisions

The reviewers have sent their reviews and minor changes are still needed.

Reviewer 1 ·

Basic reporting

a. I don’t think figure 1 (from the original submission) should have been removed. It provided a very nice visualization of the results – it just needed to be cleaned up to improve the appearance. Removing it altogether makes the paper worse off.
b. Notes on table 1 explaining the p column need to be clearer
c. There is a lot of inconsistency in using either “experimental or control” groups or using “stroboscopic and non-stroboscopic” groups. This needs to be amended.

Experimental design

a. Thanks for outlining the reasons behind the choice of the flicker frequency test – it was very informative and logical. I think that the readers would also benefit from some of this justification – just a line or two summarising the key point and including the most relevant citations.

Validity of the findings

a. Original comments have been adequately addressed.

Additional comments

a. Original comments have been adequately addressed.

·

Basic reporting

Improved based on recommendations

Experimental design

Improved based on recommendations

Validity of the findings

Improved based on recommendations

Additional comments

I would like to thank the authors for their careful consideration off the comments and the changes made to the manuscript. There are only few minor points left, that I would like to address.
1) The title states “stroboscopic protocols” which indicates that different stroboscopic protocols were compared. This is misleading since there is one intervention and one control group. Therefore, I would suggest to rephrase it to “stroboscopic training”. This would also be more consistent with the wording in the abstract “…effects of in-situ stroboscopic training on visual,…”.
2) L. 425: The term “stroboscopic patterns” is not clear although I would assume that the authors mean stroboscopic training. My suggestion would be to go with “stroboscopic training” or to avoid redundancy “stroboscopic intervention” for better clarity. Same in line 429. Across the manuscript the wording changes between “stroboscopic training”, “stroboscopic pattern” or “Stroboscopic protocol conditions”. The terms should be more consistent.
3) Ll478-480: The statement is too strong in my opinion, since the study of Sasada et al. does not show different adaptations as it is a cross-sectional study (as it stands it sounds like an interventional study). Rather the results indicate that athletes required to quickly respond to moving stimuli rely more on their dorsal stream. This should be clarified to avoid misunderstandings.

---

## Round 0.3 · accepted · Accept

I have now had the opportunity to read your revised manuscript and your responses to the reviewers' comments. I believe that you have addressed the concerns raised, and I am happy to accept your manuscript.